# The Usage of Antibiotics by COVID-19 Patients with Comorbidities: The Risk of Increased Antimicrobial Resistance

**DOI:** 10.3390/antibiotics11010035

**Published:** 2021-12-29

**Authors:** Basit Zeshan, Mohmed Isaqali Karobari, Nadia Afzal, Amer Siddiq, Sakeenabi Basha, Syed Nahid Basheer, Syed Wali Peeran, Mohammed Mustafa, Nur Hardy A. Daud, Naveed Ahmed, Chan Yean Yean, Tahir Yusuf Noorani

**Affiliations:** 1Department of Microbiology, Faculty of Life Sciences, University of Central Punjab, Lahore 540000, Pakistan; dr.basitzeshan@ucp.edu.pk (B.Z.); naveed.malik@student.usm.my (N.A.); 2Conservative Dentistry Unit, School of Dental Sciences, Universiti Sains Malaysia, Health Campus, Kubang Kerian, Kota Bharu 16150, Kelantan, Malaysia; tahir@usm.my; 3Department of Conservative Dentistry & Endodontics, Saveetha Dental College & Hospitals, Saveetha Institute of Medical and Technical Sciences University, Chennai 600077, Tamil Nadu, India; 4Department of Restorative Dentistry & Endodontics, Faculty of Dentistry, University of Puthisastra, Phnom Penh 12211, Cambodia; 5Basic Health Unit Hospital (BHU) Mora, Tehsil and District Nankana Sahib, Nankana Sahib 39100, Pakistan; nadia.afzal511@gmail.com; 6Faculty of Medicine, Riphah International University, Islamabad 46000, Pakistan; 5400@students.riphah.edu.pk; 7Department of Community Dentistry, Faculty of Dentistry, Taif University, P.O. Box 11099, Taif 21944, Saudi Arabia; sakeena@tudent.edu.sa; 8Department of Restorative Dental Sciences, College of Dentistry, Jazan University, Jazan 45142, Saudi Arabia; syednahidbasheer@gmail.com; 9Department of Periodontics, Armed Forces Hospital Jizan, Jazan 82722, Saudi Arabia; doctorsyedwali@yahoo.in; 10Department of Conservative Dental Sciences, College of Dentistry, Prince Sattam Bin Abdulaziz University, P.O. Box 173, Al-Kharj 11942, Saudi Arabia; ma.mustafa@psau.edu.sa; 11Faculty of Sustainable Agriculture, Universiti Malaysia Sabah, Sandakan Campus, Locked Bag No.3, Sandakan 90509, Sabah, Malaysia; nur.hardy@ums.edu.my; 12Department of Medical Microbiology and Parasitology, School of Medical Sciences, Universiti Sains Malaysia, Kubang Kerian, Kota Bharu 16150, Kelantan, Malaysia; yychan@usm.my

**Keywords:** antibiotic susceptibility, antimicrobial resistance pattern, antimicrobial stewardship, comorbidity, COVID-19

## Abstract

Antimicrobial resistance (AMR) is a global health issue that plays a significant role in morbidity and mortality, especially in immunocompromised patients. It also becomes a serious threat to the successful treatment of many bacterial infections. The widespread and irrelevant use of antibiotics in hospitals and local clinics is the leading cause of AMR. Under this scenario, the study was conducted in a tertiary care hospital in Lahore, Pakistan, from 2 August 2021 to 31 October 2021 to discover the prevalence of bacterial infections and AMR rates in COVID-19 patients admitted in surgical intensive care units (SICUs). Clinical samples were collected from the patients and we proceeded to identify bacterial isolates, followed by antibiotic susceptibility testing (AST) using the Kirby Bauer disk diffusion method and minimum inhibitory concentration (MIC). The data of other comorbidities were also collected from the patient’s medical record. The current study showed that the most common pathogens were *E. coli* (32%) and *Klebsiella pneumoniae* (17%). Most *E. coli* were resistant to ciprofloxacin (16.8%) and ampicillin (19.8%). *Klebsiella pneumoniae* were more resistant to ampicillin (13.3%) and amoxycillin (12.0%). The most common comorbidity was chronic kidney disease (CKD) and urinary tract infections (UTIs). Around 17 different types of antibiotic, the carbapenem, fluoroquinolones, aminoglycoside, and quinolones, were highly prevalent in ICU patients. The current study provides valuable data on the clinical implication of antibiotics consumed by COVID-19 patients in SICUs and the AMR rates, especially with different comorbidities.

## 1. Introduction

The pandemic of coronavirus disease 2019 (COVID-19) and antimicrobial resistance (AMR) are two simultaneous and interacting health concerns that provide significant learning opportunities. They may interact because, given the lack of particular therapies, there is a desire to employ current antimicrobials to treat critically ill COVID-19 patients [1]. The COVID-19 pandemic helps to illustrate the possible long-term effects of AMR, which is less severe but not less critical since their measurements and outcomes are comparable. Understanding how COVID-19 influences AMR trends and what to assume if they remain the same or increase in AMR will help us plan the next steps in addressing AMR [2].

COVID-19 infections have far exceeded bacterial co-infection and mortality rates compared to other common respiratory viral infections [2]. The co-infection of SARS-CoV-2 with other microbes, mainly bacteria and fungus, is a determining factor in COVID-19 development, making diagnosis, treatment, and prognosis more complicated. In individuals with COVID-19, bacterial co-infection has been linked to disease progression and prognosis. This scenario increases the need for critical care units, antibiotic therapy, and mortality [3]. Unfortunately, due to their widespread use, we may face the emergence of multi-drug resistant (MDR) pathogens leading to reduced efficacy of most potent antimicrobials [3,4]. AMR is a global problem that poses a severe threat to the success of treating a wide range of bacterial infections and affects many hospitalised patients, and most probably becomes a serious threat to the patients who are admitted to the SICUs [5,6].

Overall, the selection and development of highly drug-resistant bacteria due to the increased use of antibiotics and disinfectants may impact the clinical prognosis of severe COVID-19 patients receiving emergency hospital care, resulting in poor patient outcomes [7,8]. In this context, highly and extensively drug-resistant organisms have been documented to cause significant co-infections in COVID-19 patients, and mortality has recently been recorded in situations when bacterial co-infections were reported in COVID-19 patients [9,10].

The bacterial co-infections that arise during SARS-CoV-2 infection must be identified and characterised in a timely fashion [11,12]. Several studies have looked into the prevalence of bacterial co-infections in COVID-19 patients, finding highly heterogeneous distributions (with differences of >50%) that can be attributed to clinical and epidemiological characteristics of each geographical location, as well as diagnostic methods and criteria, used [13,14,15]. As a result, a study of hospitalised patients might increase our understanding of how viruses and bacteria interact during severe disease and give detailed information on COVID-19 in our environment. Similarly, identifying the main sociodemographic and clinical factors linked to bacterial co-infection in COVID-19 patients is critical to prioritising potential risk groups and institutional clinical and epidemiological surveillance programs to guide future etiological studies. Keeping in view the threat of high AMR rates in the COVID-19 pandemic, the current study was conducted to discover the prevalence of bacterial infections and AMR rates and other comorbidities with mortality rates in COVID-19 patients who were admitted to SICUs.

## 2. Materials and Methods

### 2.1. Ethical Consideration

Before starting the research, ethical approval was obtained from the Human Research Ethics Committee, University of Central Punjab, Lahore, Pakistan. Before proceeding with the sample collection, written informed consent was obtained from the patient (or from the dependent if the patient was unstable) before proceeding with the sample collection. Studied patients have been followed up for the consumption of antibiotics from the day of admission to the day of discharge (recovery or death). The demographic data (age, gender), comorbidities, recommended antibacterial drugs, the total number of given antibacterial drugs, and the brand names were recorded in a pre-defined data collection sheet.

### 2.2. Sample Collection

The current study was conducted by the Department of Microbiology, Faculty of Life Sciences, University of Central Punjab, Lahore, collaborating with a tertiary care hospital in Lahore, Pakistan, from 2 August 2021 to 31 October 2021. A total of 856 COVID-19 positive patients (admitted in SICUs) were recruited for the current study. To include more patients, only one type of sample was collected from each patient. The respiratory samples including sputum (*n* = 165), tracheal aspirate (*n* = 156), bronchoalveolar lavage (*n* = 117), and pleural fluid (*n* = 3) were collected with other samples including urine (*n* = 238), wound swab (*n* = 102), blood (*n* = 60), Foley’s catheter tip (*n* = 6), pus swab (*n* = 6), and abscess (*n* = 3) to proceed further for bacterial cultures. The samples were collected under sterile conditions and strict standard operating procedures (SOPs). After sample collection, these were immediately transported to the Microbiology laboratory at the University of Central Punjab, Lahore, for further processing.

### 2.3. Isolation and Identification of Bacterial Isolates

All samples except urine were proceeded for Gram staining first to see the microscopic characteristics of bacteria and then, based on the sample type, inoculated (including urine samples) on blood, MacConkey, chocolate and cysteine electrolyte deficient (CLED) agar media. After culture inoculation, the agar plates were incubated at 37 °C initially for 18 to 24 h. After first incubation, the agar plates were observed for the appearance of bacterial colonies. In case if there were no bacterial colonies, the plates were re-incubated for the second and third reading, respectively, except for blood cultures which were reinoculated and rechecked until the seventh day of sample collection. The negative plates were reported as “No bacterial growth” while the positive bacterial cultures further proceeded for bacterial identification and AST.

The final bacterial identification was undertaken using Gram staining, the appearance of bacterial colonies on the agar plates and biochemical identification. The colonies of Gram-positive bacteria proceeded to the catalase test first, and if the catalase test was positive, the colonies proceeded further for coagulase, DNAs, and Optochin disk tests, accordingly. However, the Gram-negative bacteria were first examined in terms of their appearance as lactose fermenters or non-fermenter and then proceeded for indole, citrate, and oxidase tests and the analytical profile indexing (API) biochemical identification kit.

### 2.4. Antibiotic Susceptibility Testing (AST)

After the isolation and identification of the organisms, the AST was performed to check their antibiotic susceptibility patterns using a Kirby Bauer disk diffusion assay and MIC (where applicable) [16]. The bacterial colonies were first mixed in a pre-prepared 0.5% MacFarland standard solution to dilute colonies to check the susceptibility patterns. After the inoculum of the test organism, these were inoculated three-dimensionally into a Muller Hinton (MH) agar (MH) agar plate with the help of a sterile cotton swab.

The antibiotics were checked according to the clinical laboratory standards institute (CLSI) guidelines (2017) for each microorganism [16]. The antibiotics were dispensed on inoculated MH agar plates and incubated for 18 to 24 h at 37 °C. After the incubation period, the zone of inhibition was measured using a labelled measuring scale. The CLSI guidelines followed the zone of inhibition (measured in mm) for each of the antibiotic v/s organisms. The final AST results were noted as resistant, sensitive or intermediate.

According to the CLSI guidelines (2017), the zone of inhibition for fosfomycin was only available to be reported in *E. coli* and *Enterococcus faecalis* isolates using the disk diffusion method. The sensitivity patterns were based on minimum inhibitory concentrations (MICs) for all other recommended bacterial isolates.

### 2.5. Statistical Analysis

The final data was recorded in Microsoft Excel and transferred to the SPSS version 26.0 (IBM, New York, NY, USA). The Frequencies and percentages were calculated from the recorded data. The association between the comorbidities and treatment outcome, type of bacterial infection, and severity of COVID-19 were analysed using the Pearson Chi-square (X2) test where *p*-value of <0.05 was considered statistically significant.

## 3. Results

### 3.1. Distribution of Clinical Bacterial Isolates among Coronavirus Disease 2019 (COVID-19) Patients

From the total 856 patient samples, *n* = 506 (59.11%) were male and *n* = 350 (40.88%) were females as shown in Table 1. A total of 342 (39.95%) samples were found to be positive for different bacterial isolates. Most of the bacterial cultures were found to be positive in urine (*n* = 136, 39.76%) and tracheal aspirate cultures (*n* = 51, 14.91%) followed by broncho-alveolar lavage (*n* = 48, 14.03%), wound swab (*n* = 37, 10.81%), blood (*n* = 32, 9.35%), sputum (*n* = 24, 7.01%), pus swab (*n* = 5, 1.46%), Foley’s catheter tip (*n* = 4, 1.16%), abscess (*n* = 3, 0.87%), and pleural fluid (*n* = 2, 0.58%).

### 3.2. Clinical Isolates from Various Infections

The most common diagnosed comorbidity was chronic kidney disease (CKD), along with urinary tract infection (UTI) (21.4%), which was caused mainly by *E. coli* (50%), followed by the *Klebsiella pneumoniae* (20%). The second most diagnosed comorbidity was sepsis and hematuria (19.1%), also caused mainly by *E. coli* (52.9%), followed by *Klebsiella pneumoniae* (11.8%) and *Proteus* spp. (11.7%), respectively. The least common diagnosis was respiratory tract infections (1.11%), caused mainly by *Klebsiella pneumoniae* (100%). The BSI followed by the lung abscesses (1.07%) was caused mainly by the *Klebsiella pneumoniae*. (100%). The prevalence of bacterial isolates is shown in Table 2. Appendix A: Appendix A shows the beta hemolysis by *Streptococcus* spp. On blood agar after 24 h of incubation at 37 °C. Appendix A shows the appearance of Gram-negative rods under observation at the 100× lens of the microscope. At the same time, Appendix A shows the appearance of Gram-positive cocci. Appendix A shows the positive bile esculin test, which is for *Enterococcus faecalis*. Appendix A shows the positive results for *Escherichia coli* using the API kit.

### 3.3. Antibiotic Susceptibility Patterns of Clinical Isolates

The antibiotic resistance patterns of individual bacterial isolates are shown in Figure 1, Figure 2, Figure 3 and Figure 4.

In SICUs, different antibiotics were prescribed and given to patients with a different diagnosis, as shown in Table 3.

### 3.4. Distribution of COVID-19 Patients with Major Comorbidities

Apart from COVID-19, 36.09% (*n* = 309) of patients also suffered from bacterial pneumoniae, UTIs, meningoencephalitis and sepsis. The patients with pneumoniae were also diagnosed with aspiration, diabetes mellitus (DM), chronic obstructive pulmonary disease (COPD), ischemic heart disease (IHD) and hypertension (HTN). The patients with UTIs, meningoencephalitis and sepsis were also diagnosed with other comorbidities, which were the main reason for mortality, as shown in Table 4.

A significant association (*p* < 0.001) was found between the COVID-19 patients, comorbidities and the mortality rate. The correlation between co-infections and COVID-19 severity has been shown in Table 5.

### 3.5. Mortality and Recovery Rate of the Patient

The recovery and death rate among the COVID-19 patients was recorded during their stay in the hospital. From the total 856 COVID-19 patients, the recovery rate was 76.16% (*n* = 652). The death rate was 23.83% (*n* = 204). The most common death rate among COVID-19 patients with different comorbidities was the CLD cases, followed by pneumoniae, UTIs, sepsis, cellulitis, pancreatic cancer, ascites, CKD, pyelonephritis rigours, cystitis and hematuria.

## 4. Discussion

Antibiotics are the most commonly prescribed drugs among hospitalised patients, especially in SICUs. It is imperative to use the appropriate antibiotics in intensive care units with few prescriptions as an acceptable quality of care, infection control, cost reduction, and length of hospital stay [17,18,19]. Patients admitted in the SICU are critically ill requiring prescribed the medicine without waiting for the culture reports that give information about the antimicrobial resistance pattern of the suspected organism for a specific cause [20,21]. In terms of the culture reports, there is no possibility to wait for reports because these take a minimum of 48 h, and as a result antibiotic resistance occurs in patients admitted to the SICUs [22]. The current study was conducted among COVID-19 patients admitted in SICUs of tertiary care hospitals, requiring monitoring and special care to analyse the antibiotics utilisation pattern and determine the prevalence of AMR. Our analysis included each organism’s antibiotics sensitivity and resistance pattern isolated from patients admitted in SICUs.

A previous study on microbial infection and antibiotic resistance patterns in COVID-19 patients admitted in SICUs of tertiary care hospitals showed that *Pseudomonas* was the most common organism identified in the medical ICU, followed by *Klebsiella pneumonia* [23]. A study on the prevalence of microorganisms and bacterial resistance in the SICU of the Bangabandhu Sheikh Mujib Medical University of Bangladesh showed that the maximum identified organism was *Acetobacter*. (45.4%), of *P. aeruginosa* (32.2%), *Proteus* (11%)*, Klebsiella pneumoniae* 10%, and *E. coli* (3%) were identified [24]. A study by Mehta et al., 2015 on ICU patients revealed that the *Pseudomonas* spp. (29.1%) was the most common organism, followed by *Acinetobacter* spp. (27.5%) [18]. Another previous analysis of AST and bacteriology profile on patients at tertiary care hospitals in Ahmadabad showed that *Acinetobacter* spp. [30.9%] was the most common organism, after coming *Klebsiella* spp. (29.8%) and *P. aeruginosa* (22.9%) [19]. However, in the current study, the most common isolated organism was *E. coli* (38%), followed by *Klebsiella pneumoniae* (24%), *P. aeruginosa* (14%). While *Streptococcus agalactiae*, *Citrobacter freundii*, *Serratia liqeuficiens*, and *Stenotrophomonas maltophilla* were 1.7%, 1.1%, 1.1% and 1.4%, respectively.

In Jordan, a study was conducted to see the AMR rates, the prevalence of bacterial infections, and misuse of antibiotics. They conducted a study to resolve the resistance rate of Gram-negative bacteria in patients admitted to the ICU of Prince Hasen hospital. It revealed that *P. aeruginosa* was the most resistant pathogen and broad-spectrum antibiotic-resistant bacteria [15]. A similar previous study on the uropathogens of ICU showed that *E. coli* was highly susceptible to imipenem, meropenem, and nitrofurantoin [25]. The second most common organism in the current study was *Klebsiella pneumoniae*, 100% resistant to ampicillin and 91% to Amp-clavulanic acid. For *Klebsiella pneumoniae*, the treatment option was amikacin and gentamicin. Work on the AMR pattern of the bacterial pathogen in the ICU of Fatimawati hospital showed high resistance to cephalexin (96.3%), cefotaxime (64.3%), and ceftriaxone (61.0%) shown by *P. aeruginosa*. The most effective antibiotic against *P. aeruginosa* was amikacin after imipenem (81.3%) and meropenem (75.2%). *Klebsiella pneumoniae* showed resistance to cephalexin, ceftriaxone, ceftazidime (86.6%) (75.9%) (73.4%) respectively [26]. In the current study, the most common organism, *E. coli,* showed ampicillin resistance (83%) and Amp-clavulanic acid (87%). In that case, the most effective antibiotic was imipenem and meropenem. Most *E. coli* showed high sensitivity to the imipenem and meropenem (96%). Imipenem and meropenem are usually used against both GPIs and GNIs. However, a study conducted in India showed that *Klebsiella* was more susceptible to imipenem and nitrofurantoin and showed more resistance to penicillin and gentamicin [18].

Most of the Gram-positive organisms in the current study were resistant to penicillin and tetracycline. For the treatment of the Gram-positive bacteria, linezolid, gentamicin, and vancomycin were used. Most Gram-negative organisms were resistant to two or more alarming antibiotics and shortly caused high mortality and morbidity. This will also affect the management of Gram-negative bacteria. Around 17 different types of antibiotics were used in the ICU. Carbapenem, fluoroquinolones, aminoglycoside, and quinolones were highly consumed among all the antibiotics. In a study in India, the high consumption of beta-lactam, carbapenem, metronidazole was observed. Another study in Nepal showed that ampicillin, metronidazole, and amoxicillin were the most common antibiotics [27]. Another research study showed that beta-lactam, nitroimidazoles, and fluoroquinolones were commonly used in an ICU [12]. The prevalence of MRSA in the current study was 16%, VRE was 11%, and CRE 17%, showing a high rate of AMR against most effective drugs. These high rates of AMR in SICU patients are an alarming situation for all of the clinicians and researchers and the time to take some actions to ensure the correct usage of antibiotics.

Results of the current study showed that approximately 36% of COVID-19 patients had pneumonia followed by aspiration, diabetes mellitus (DM), chronic obstructive pulmonary disease (COPD), ischemic heart disease (IHD) and hypertension. UTIs, meningoencephalitis and sepsis followed by different comorbidities were also diagnosed. COVID-19 with bacterial pneumonia were the most important cause of distress and mortality. DM, HTN, CKD were significant comorbidities among ICU patients. Many patients had DM and HTN at the same time. Females experienced more complications and comorbidities than males. However, a previous systematic review and meta-analysis by Yang et al. (2020) reported that HTN and diabetes were the most common comorbidities, followed by cardiovascular diseases (CVD) and pneumonia. The pooled odds ratio of HTN, pneumonia and CVD in severe patients was lower than in non-severe patients [28]. We looked at the incidence of comorbidities in COVID-19 infection patients and discovered that underlying conditions, such as HTN, pneumonia, DM, CKD and COPD, might be a risk factor for high AMR rates among COVID-19 patients.

The current study was performed in a single hospital, and only those patients recruited for the current study were admitted in the SICUs. The sample size was smaller due to the single centre/unit study. Furthermore, multicentral studies with a larger sample size are recommended.

## 5. Conclusions

The current study points to a significant prevalence of bacterial infections and high AMR rates in COVID-19 patients, especially with specific comorbidities and complications, making it challenging to identify the priority of treatment groups and improve the care of these infections. These high rates of AMR may lead to high mortality rates; that is why a specific antibacterial usage policy of institutions and SICUs are needed to control this problem. Periodic audits are required to monitor compliance with these guidelines. In a hospital setup, antibiotic sensitivity and AMR patterns must be considered to direct the infection control expert or clinician to begin empirical antibiotics in severe cases. An intensive and systematic effort is needed to quickly classify high-risk patients and minimise the irrelevant use of antibiotics, one of the primary reasons for increased AMR rates in developing countries.

## Figures and Tables

**Figure 1 antibiotics-11-00035-f001:**
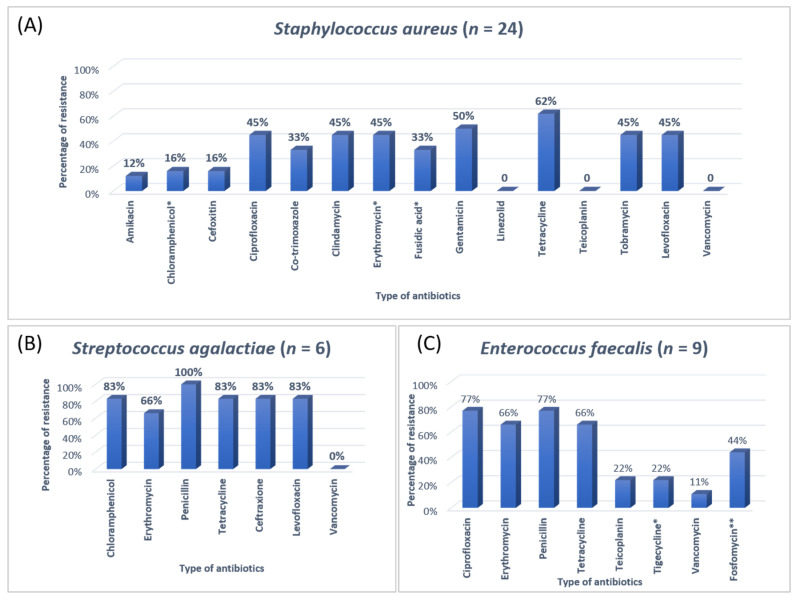
The antibiotic resistance patterns in (**A**) *Staphylococcus aureus*, (**B**) *Streptococcus agalactiae*, (**C**) *Enterococcus faecalis*. * Not reported in urinary isolates. ** Reported in urinary isolates only.

**Figure 2 antibiotics-11-00035-f002:**
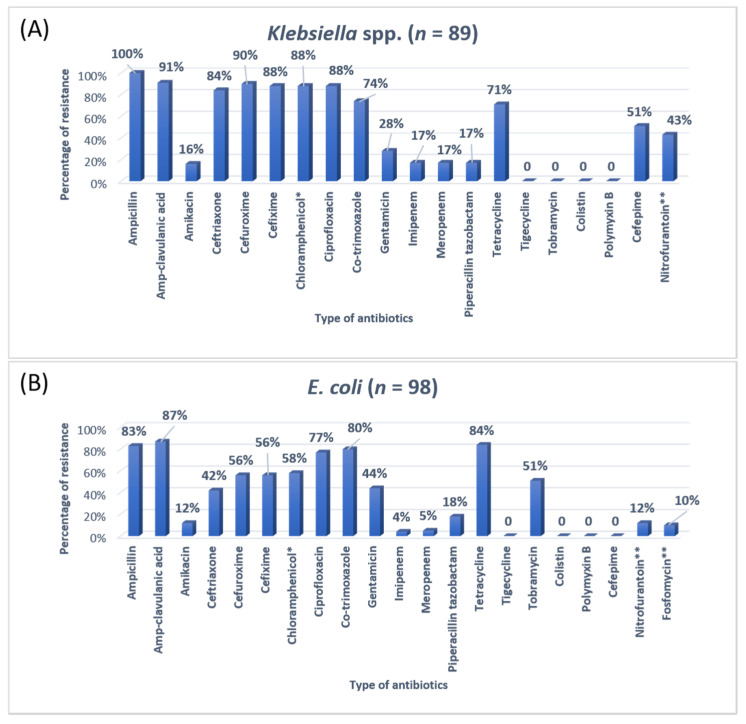
The antibiotic resistance patterns in (**A**) *Klebsiella* spp, (**B**) *Escherichia coli* * Not reported in urinary isolates. ** Only reported in urinary isolates.

**Figure 3 antibiotics-11-00035-f003:**
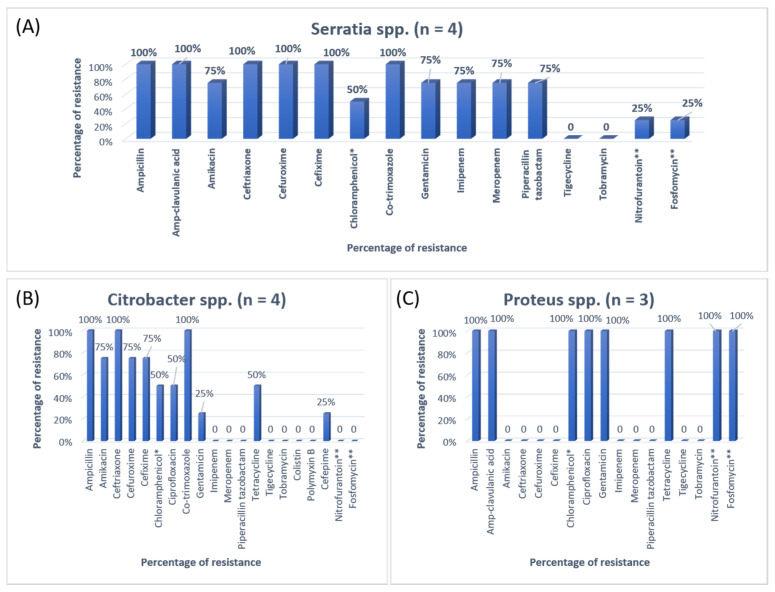
The antibiotic resistance patterns in (**A**) *Serratia* spp., (**B**) *Citrobacter* spp., (**C**) *Proteus* spp. * Not reported in urinary isolates. ** Only reported in urinary isolates.

**Figure 4 antibiotics-11-00035-f004:**
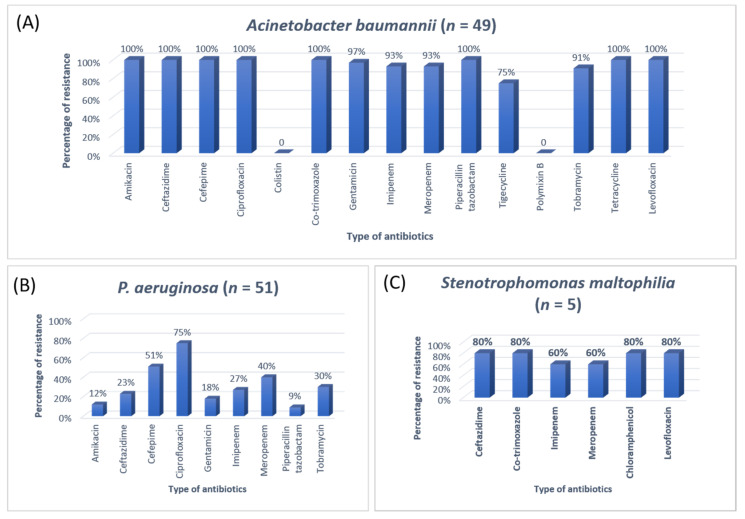
The antibiotic resistance patterns in (**A**) Acinetobacter baumannii, (**B**) Pseudomonas aeruginosa, (**C**) Stenotrophomonas maltophilia.

**Table 1 antibiotics-11-00035-t001:** Gender-wise and age group-wise distribution of coronavirus disease 2019 (COVID-19) patients in intensive care unit (ICU).

Age Group (Years)	Gender		Total
	Male	Female	
01–20	72	74	146
21–40	99	102	201
41–60	132	76	208
61–80	181	86	267
>80	22	12	34
Total	506	350	856

**Table 2 antibiotics-11-00035-t002:** The general prevalence of bacterial co-infections in COVID-19 patients admitted in ICU.

Bacterial Isolates		*n*	%
Gram-Positive bacteria (*n* = 39, 11.40%)			
*Staphylococcus aureus*	MRSA	4	1.16
	MSSA	20	5.84
*Enterococcus faecalis*	Non-VRE	8	2.33
	VRE	1	0.29
*Streptococcus agalactiae*		6	1.75
**Gram-Negative bacteria (*n* = 303, 88.59%)**			
*Escherichia coli*	Non-CRE	93	27.19
	CRE	5	1.46
*Klebsiella pneumoniae*	Non-CRE	68	19.88
	CRE	16	4.67
*Pseudomonas aeruginosa*		51	14.91
*Acinetobacter baumannii*		49	14.32
*Stenotrophomonas maltophilia*		5	1.46
*Klebsiella oxytoca*		5	1.46
*Citrobacter freundii*		4	1.16
*Serratia liquefaciens*		4	1.16
*Proteus vulgaris*		3	0.87

MRSA: Methicillin-resistant *S. aureus*. MSSA: Methicillin sensitive *S. aureus*. VRE: Vancomycin resistant *Enterobacteriaceae.* CRE: Carbapenem resistant *Enterobacteriaceae*.

**Table 3 antibiotics-11-00035-t003:** Clinical isolates from various samples and their prescribed antibiotics.

Variables		Drug Administration	Number (*n*)	Percentage (%)
Type of antibiotic used	Vancomycin	IV	23	6.72
	Teicoplanin	IV	2	0.58
	Linezolid	IV	4	1.16
	Imipenem	IV	29	8.47
	Meropenem	IV	27	7.89
	Amikacin	IV	54	15.78
	Gentamicin	IV/Oral	36	10.52
	Tobramycin	IV/Oral	23	6.72
	Nitrofurantoin	IV/Oral	76	22.22
	Fosfomycin	IV/Oral	84	24.56
	Piperacillin-tazobactam	IV	62	18.12
	Colistin	IV	21	6.14
	Polymyxin B	IV	16	1.75
	Ciprofloxacin	IV/Oral	73	21.34
	Ceftriaxone	IV	34	9.94
	Cefepime	IV	23	6.72
Combination of antibiotics	One		229	66.95
	Multiple (two or more)		113	33.04
Days of antibiotic treatment	1 to 7 days		204	59.64
	8 to 14 days		91	26.60
	>15 days		47	18.71

IV: Intravenous.

**Table 4 antibiotics-11-00035-t004:** Distribution of comorbidities among COVID-19 patients admitted in the intensive care unit and their outcome.

Sr. No	Comorbidities	Total Number of Patients	Outcome	*p*-Value
1.	Pneumonia, Aspiration, DM, HTN, IHD, COPD, CLD	135	64 Recovered 71 Died	<0.001
2.	UTIs, Dementia, DM, HTN, CKD	140	121 Recovered 19 Died	
3.	Meningoencephalitis, Parkinsonism, HTN, Stroke	2	2 Recovered	
4.	Sepsis, DM, GI disorders	32	29 Recovered 3 Died	

CKD: Chronic kidney diseases. CLD: Chronic liver disease. GI: Gastrointestinal.

**Table 5 antibiotics-11-00035-t005:** Distribution of co-infections among COVID-19 patients admitted in SICUs.

Co-Infections	COVID-19 Severity			*p*-Value
	Mild	Moderate	Severe	
Upper respiratory tract infections	4	5	15	0.028
Lower respiratory tract infections	13	21	67	
Bacteremia	3	8	21	
Gastrointestinal infections	4	23	18	
Urinary tract infections	11	41	88	

## Data Availability

The data will be shared upon a reasonable request to the corresponding author.

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
