# Peer review of "The Usage of Antibiotics by COVID-19 Patients with Comorbidities: The Risk of Increased Antimicrobial Resistance"

_antibiotics, 2021, doi:10.3390/antibiotics11010035_

Round 1

Reviewer 1 Report

The revised manuscript has been improved while there are still some issues that have not been addressed yet. Here are some remarks to the revised manuscript:
1. Statistical analysis: “The data were further analysed for statistical analysis using the Chi-Square test.” Why do you prefer the Chi-Square test in this work? In my experience, it is not a good analysis method for this study.
2. I previously pointed out that this work only involved a comparatively small number of patients. Nevertheless, the authors did not update this limitation in the revision. 
3. The manuscript contains 8 Tables but none Figure. It should be noted that Figure and Histogram are more intuitive and will help improve readability. I recommend transferring Tables 3, 4 and 5 into Figures.

Author Response

Response to Reviewer 1 Comments

The revised manuscript has been improved while there are still some issues that have not been addressed yet. Here are some remarks to the revised manuscript:

Response: We want to thank the academic editor and the reviewers for taking their precious time to review this manuscript and give us their comments. We would like to explicitly state that we agree with all the comments as these helped us improve the quality of our paper. We have made a conscious effort to answer all the remarks in the paper as advised by the reviewers and highlighted changes made in yellow for their convenience. Kindly consider these and excuse us for any lapse on our part. Please let us know if more changes need to be made to improve the paper.

  1. Statistical analysis: "The data were further analyzed for statistical analysis using the Chi-Square test." Why do you prefer the Chi-Square test in this work? In my experience, it is not a good analysis method for this study.

Response: The one-way Anova test could not run because fewer values in the variables mild severity of COVID-19, resulting in false significance. In the variable "Severity of COVID-19", the sub-variable Mild has 3 values, and Moderate have 5 values. For one-way ANOVA, the value in each variable should be a minimum of 5. Furthermore, Fisher's exact test could not be run by SPSS because fewer variables in the data. Because of these reasons, we run the Pearson Chi-square test to check the association between bacterial infections and severity of COVID-19, comorbidities and treatment outcome.

  1. I previously pointed out that this work only involved a comparatively small number of patients. Nevertheless, the authors did not update this limitation in the revision.

Response: Thank you for your insightful suggestion; the study's limitations have been added in the last discussion section in the revised manuscript as per the comments.

"The current study was performed in a single hospital, and only those patients were recruited for the current study who were admitted in the SICUs. The sample size was less due to the single centre/unit study. Further, multicentral studies with larger sample size are recommended."

  1. The manuscript contains 8 Tables but none Figure. It should be noted that Figure and Histogram are more intuitive and will help improve readability. I recommend transferring Tables 3, 4 and 5 into Figures.

Response: Thank you for your insightful suggestion; Tables 3, 4, and 5 has been transferred to Figures 1,2,3 and 4 as per your suggestions and comments.

Reviewer 2 Report

Dear authors

thank you for your efforts to improve the quality and the scientific soundness of your work. Here my comments for suggestions of revision:

-in the abstract and in the sample collection paragraph the timeline of the study should be specified: beginning and end of study data collection

-Paragraph 2.4: how was fosfomycin susceptibility evaluated given the peculiarity? Please specify in the re

-paragraph 3.1 please distinguish patients from samples, and clarify if for each patient there was one only sample.

Line 154: "A total of 342 (39.95%) samples were found positive for different bacterial isolates". This sentence is confusing: how many samples for each patient? 

Paragraph 3.3: please specify when talking of fosfomycin if talking of iv or oral formulation (Table 6), it makes a big difference both for Gram pos and Gram neg.

Line 193: how was correlation studied? Please specify in the Statistical Method section

Line 224-233: please reduce length of this paragraph citing main points.

Could author express antibiotics usage during the study period in DDD and compare those rates with a pre-covid period? It would add great interest to the study.

Author Response

Response to Reviewer 2 Comments

Thank you for your efforts to improve the quality and the scientific soundness of your work. Here my comments for suggestions of revision:

Response: We want to thank the academic editor and the reviewers for taking their precious time to review this manuscript and give us their comments. We would like to explicitly state that we agree with all the comments as these helped us improve the quality of our paper. We have made a conscious effort to answer all the remarks in the paper as advised by the reviewers and highlighted changes made in yellow for their convenience. Kindly consider these and excuse us for any lapse on our part. Please let us know if more changes need to be made to

improve the paper.

-in the abstract and in the sample collection paragraph the timeline of the study should be specified: beginning and end of study data collection

Response: Thank you for your insightful suggestion and comments; the sample collection timeline has been specified in the revised manuscript in the abstract (August 2021 to October 2021).

-Paragraph 2.4: how was fosfomycin susceptibility evaluated given the peculiarity? Please specify in the re

Response: Thank you for your insightful suggestion and comments. The following sentence has been added in the revised version of the manuscript:

"According to CLSI guidelines (2017), the zone of inhibition for fosfomycin was only available to be reported in E. coli and Enterococcus faecalis isolates. For all other recommended bacterial isolates, the sensitivity patterns were based on minimum inhibitory concentrations (MICs)."

-paragraph 3.1 please distinguish patients from samples, and clarify if for each patient there was one only sample.

Response: Thank you for your insightful suggestion and comments. The following sentences have been added in the revised version of the manuscript:

"From the total 856 patient samples, n = 506 (59.11%) were male, and n = 350 (40.88%) were females as shown in Table 1."

"To include more patients, from each patient, only one type of sample was collected."

Line 154: "A total of 342 (39.95%) samples were found positive for different bacterial isolates". This sentence is confusing: how many samples for each patient?

Response: Thank you for your insightful suggestion and comments. The comment have been addressed in the material and method section of the revised manuscript. "To include more patients, from each patient, only one type of sample was collected."

Paragraph 3.3: please specify when talking of fosfomycin if talking of iv or oral formulation (Table 6), it makes a big difference both for Gram pos and Gram neg.

Response: Thank you for your insightful suggestion and comments. A new column for the route of drug administration has been added in table 3.

Line 193: how was correlation studied? Please specify in the Statistical Method section

Response: Thank you for your insightful suggestion and comments. The following sentence has been added in the revised version of the manuscript.

"The association between the comorbidities and treatment outcome, type of bacterial infection and severity of covid- 19 were analyzed using Pearson Chi-square (X2) test where P-value of < 0.05 was considered as statistically significant."

Line 224-233: please reduce length of this paragraph citing main points.

Response: Thank you for your insightful suggestion and comments. The length of paragraphs has been reduced accordingly in the revised version of the manuscript.

Could author express antibiotics usage during the study period in DDD and compare those rates with a pre-covid period? It would add great interest to the study.

Response: Thank you for your insightful suggestion and comments. A new column for the route of drug administration has been added in table 3. We highly appreciate your suggestion, but the hospital unit did not disclose the dosage of the drug delivered. Hence, we request you to excuse us in this regard kindly.

This manuscript is a resubmission of an earlier submission. The following is a list of the peer review reports and author responses from that submission.

Round 1

Reviewer 1 Report

This manuscript provides a report on the clinical implication of antibiotics consumed by COVID-19 patients in surgical intensive care units, especially with comorbidities. Overall, the presented results are interesting and could be valuable for other researchers in this field. However, there are still some issues to be addressed.
1. What is the aim of this manuscript? It is hard to understand the objective and significance of this work from the presented introduction. Please revise it.
2. One important point has been missed: Why do you prefer the Covid-19 patients admitted in surgical intensive care unit in this manuscript? More interpretation should be added.
3. “While remining 39 (11.40%) were gram-positive bacteria. The most common bacteria were Escherichia coli (n = 98, 28.65%) and Klebsiella pneumoniae (n = 84, 24.56%) followed by Pseudomonas aeruginosa (n = 51, 14.91%), Acineto-bacter baumannii (n = 49, 14.32%), Staphylococcus aureus (n = 24, 7.01%), Enterococcus faecalis (n = 9, 2.63%), Streptococcus agalactiae (n = 6, 1.75%), Stenotrophomonas maltophilia (n = 5, 1.46%), Klebsiella oxytoca (n = 5, 1.46%), Citrobacter freundii (n = 4, 1.16%), Serratia liquefaciens (n = 4, 1.16%) and Proteus vulgaris (n = 3, 0.87%).” Please transfer this text into a Table or Figure.
4. Table 2, 3, and 4: What is the unit of the Zone of Inhibition? cm or mm?
5. Actually, this work involved a comparatively small number of patients, and I even can’t find any statistical analysis or statistical significance. Therefore, these findings reported should be considered with extreme caution.

Reviewer 2 Report

Comments

In the manuscript titled “Correlation of Antimicrobial Consumption and Resistance Among Covid-19 Patients Admitted in Surgical Intensive Care Unit (Sicu)” the authors mentioned that the work focused “to find out the correlation of antibiotic consumption with the type of microbial infection and the recovery & death of the patient admitted to SICUs.” However, there is no correlation analysis to be found throughout the manuscript. A correlation analysis goes deeper than a simple and shallow description of the studied groups. A correlation analysis is a kind of statistical analysis performed to find a causal relationship, whether causal or not, between two variables. This relationship, if exists, could be positive or negative.

Introduction

The introduction is well written; however, the main goal of the project is not mentioned nor the reasons for performing the study.

Materials and methods

In Section 2.4 the authors mention “The antibiotics were checked according to the clinical laboratory standards institute (CLSI) for each microorganism.” What was the year of editing of the CLSI guide used by the authors? Additionally, the authors forgot to reference the CLSI guide in the References section.

Section 2.5 mentions there is a Statistical analysis that is not shown in any result presented in the manuscript. The results of this statistical analysis as well as the variables presumably used to perform it are nowhere to be found.

Results

All the results are described but the information is not analyzed at all, making the manuscript just a collection of raw data. Moreover, the same data is mentioned twice, both in the text and in the tables. Figure 1 does not make sense; you cannot have a pie chart that doesn’t add up 100%

All sections must be reorganized for clarity and to avoid repeated data.

Table 1 “Gender-wise and age group-wise distribution of COVID-19 patients in ICU” presents data that is not used for the discussion of the antibiotic resistance patterns in bacteria or any other comparison.

Section 3.2 is altogether unnecessary because there the authors only repeated the method already described in section 2.4 and referred the reader to tables 2, 3, and 4.

In tables 2, 3, and 4 the columns CODE, CONCENTRATION (ug), and ZONE OF INHIBITION as this information can be consulted in the CLSI guide.

I believe that in Section 3.5 the patients’ symptoms are irrelevant for the objective. Additionally, in Table 7 the total column, the sum of male and female patients, is not correct.

Discussion

In the Discussion sections a lot of results are repeated and lack formal analysis, there is only description.

Conclusion

The authors say “Since this study was conducted on a comparatively small number of patients, it may affect our statistics to a certain extent” but the authors never showed any statistical analysis. This is an awful mistake.

Reviewer 3 Report

Dear authors

I have read with interest your paper considering a very hard and important topic. Unfortunately I think that the paper somehow missed the title "Correlation of Antimicrobial Consumption and Resistance Among Covid-19 Patients Admitted in Surgical Intensive Care Unit (Sicu)". The paper describes collection of microbiological samples from covid-19 ICU patients and susceptibility patterns, but it does not speculate (with an appropriate statistical methodology) about correlation between antibiotic consumption and antimicrobial resistance.

You wrote in the abstract:  "The current study provides valuable data on the clinical implication of antibiotics consumed by COVID-19 patients in ICU", but actually I am afraid you cannot desume this sentence from any of the article result.

So, I think this paper need a very profound and robust revision in term of methods and data collections, after which some very interesting data could emerge.